# Biostimulant Seed Coating Treatments to Improve Cover Crop Germination and Seedling Growth

**Yi Qiu [1,2,†], Masoume Amirkhani [1,*,†] 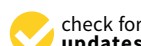, Hilary Mayton [1], Zhi Chen [2] and Alan G. Taylor [1]**

1   Cornell AgriTech, Horticulture Section, Cornell University, SIPS, Geneva, New York, NY 14850, USA; qyi0508@emails.imau.edu.cn (Y.Q.); hsm1@cornell.edu (H.M.); agt1@cornell.edu (A.G.T.)
2   College of Mechanical and Electrical Engineering, Inner Mongolia Agricultural University, Hohhot, Inner Mongolia 010018, China; sgchenzhi@imau.edu.cn
*   Correspondence: ma862@cornell.edu
†   Co-first authors.

**Abstract:** Biostimulant seed coating formulations were investigated in laboratory experiments for their potential to increase maximum germination, germination rate, germination uniformity, and seedling growth of red clover (*Trifolium pratense* L.) and perennial ryegrass (*Lolium perenne* L.) seeds. Red clover and perennial ryegrass seeds were coated with different combinations of soy flour, diatomaceous earth, micronized vermicompost, and concentrated vermicompost extract. Coated and non-coated seeds of red clover and perennial ryegrass were evaluated for germination and growth after 7 and 10 days, respectively. Red clover seed was maintained at a constant 20 °C with a 16/8 h photoperiod, whereas for perennial ryegrass seed, the germinator was maintained at 15/25 °C, with the same photoperiod as red clover. Coated treatments significantly improved germination rate and uniformity with no reduction in total germination, compared to the non-treated controls in red clover. In contrast, for perennial ryegrass, the total germination percentage of all coated seeds was reduced and displayed a delayed germination rate, compared with the non-treated controls. Shoot length, seedling vigor index, and dry weight of seedlings of coated seed treatments of both crops were significantly higher when compared to controls for both species. In addition to growth metrics, specific surface mechanical properties related to seed coating quality of seeds of both species were evaluated. Increasing the proportion of soy flour as a seed treatment binder in the coating blend increased the integrity and compressive strength of coated seeds, and the time for coatings to disintegrate. These data show that seed coating technologies incorporating nutritional materials and biostimulants can enhance seedling growth and have the potential to facilitate the establishment of cover crops in agriculture and land reclamation.

**Keywords:** seed coating; cover crop; vermicompost; biostimulant; growth enhancement

## 1. Introduction

Exponential growth in human global population, from 1.7 billion in 1900 to approximately 7.6 billion in 2019, has led to the over use and degradation of agricultural landscapes, including grasslands used for grazing, forage, and food production [1,2]. The rapid growth of populations in pastoral areas, including Inner Mongolia, China, have caused intensive overutilization of grasslands. Approximately, 40% of land area in China is classified as grassland and accounts for 13% of the world's grassland [2,3]. Overgrazing and conversion of grassland to cropland has led to declines in overall agricultural productivity due to increased soil erosion, degraded soil structure, and reduced soil fertility. Recently, China implemented vegetation restoration programs to improve biodiversity in agriculture environments, soil health, and productivity, and to reduce erosion and desertification [2]. Legumes

and ryegrasses are widely used as cover crops to reduce desertification and restore productivity on degraded grasslands [4] and are commonly used for land reclamation and restoration of abandoned mine land [5,6]. Perennial ryegrass (*Lolium perenne* L.) is a cool season grass native to southern Europe, the Middle East, Central Asia, and North Africa [7]. Ryegrass is often used to stabilize soils for erosion control and is frequently seeded with red (*Trifolium pratense* L.) or white (*Trifolium repens* L.) clovers for increased productivity in grazing and to provide nitrogen and aid in weed suppression [8]. Red clover is particularly tolerant to drought conditions and helps to improve soil structure due to its large, fast-growing (more than 60 cm/year) tap root [8]. The benefits of cover cropping in both organic and conventionally managed systems are well documented [9].

Cover crops increase soil organic matter, soil structure, nutrient retention and cycling, and reduce soil erosion [8]. However, under drought conditions, and in areas with poor soils such as arid degraded grasslands, germination and subsequent growth of cover crops are inadequate, and sowing is often unsuccessful. Seed enhancements, which can include seed priming, coating, and conditioning are frequently used to improve seed delivery during planting, and to increase seed germination, stand uniformity, seedling growth, and suppress disease [10]. Seed priming increased germination rate and overall seedling emergence in a study investigating perennial ryegrass for fall seeding under cool temperatures and improved wheat stand establishment under marginal soil conditions [11,12]. Seed treatments with fungicides and fertilizers enhanced stand establishment of perennial ryegrass in field experiments in New Zealand [13,14]. Seed enhancements via seed coating can also provide micro and macronutrients or biostimulant materials to increase germination, seedling vigor, and stand establishment [15]. Seed coating technology has been used as a promising and effective approach for enhancing establishment and yield of different grass and forage species (*Lolium perenne, Trifolium pratense, Elymus dahuricus*) in semi-arid areas of China such as Inner Mongolia [16–18].

Biostimulants are materials that can augment plant growth when applied to plants and seeds, but are not classified as fertilizers, pesticides, or soil amendments [19]. Commonly applied biostimulants include microbial inoculants, beneficial bacteria and fungi, nitrogen containing compounds, biopolymers, and plant extracts [19]. Research and use of biostimulants in agriculture has increased in recent years in an effort to reduce reliance on less sustainable conventional pesticides and fertilizers, which are often overused, in agricultural cropping systems [15,19–22]. Seed treatments require even smaller amounts of active ingredients per hectare than foliar applied treatments, primarily due to the reduced surface area treated, and increase germination and plant growth when compared to non-treated seed [23].

Modern seed coating technology utilizes different approaches depending on the shape and size of the seed and the type and amount of materials added to seeds [10,24–27]. Currently, seed pelleting, film coating, and seed encrusting are the most common coating/treatment procedures used in the seed industry to enhance plant and seedling performance. While seed pelleting, often employed to develop more uniform seeds for mechanical planting, can increase seed weight from 200 to > 5000%, film coating or encrusting utilizes much smaller quantities of materials resulting in a build-up in seed weight of between 0.5–10% and 20–200%, respectively [24]. The physical properties and thickness of the seed treatment/coating are critical factors that influence seed germination and seedling vigor. A thick hard seed coating can reduce, delay, or cause abnormal germination or may even be toxic, while a minimal, fragile seed coating can break or disintegrate before planting or not have a high enough dosage of an active ingredient to be effective. Therefore, specialized seed coating formulations must be developed and evaluated in order to be utilized effectively for any given plant species and agronomic purpose.

The specific objectives of this research were to explore plant-derived bio-based biostimulant seed coatings to enhance germination and growth of two cover crop species, red clover and perennial ryegrass, as an approach for seeding cover crops for grassland restoration. Previous research on seed coatings of broccoli and tomato with soy flour and compost materials showed promising results related to maximum germination, germination uniformity, and seedling vigor [15,25–27].

## 2. Materials and Methods

### 2.1. Seed and Coating Materials

Two species of cover crops were selected to evaluate biostimulant seed coatings in this study. Red clover 'VSN-variety not stated' seed was obtained from King's AgriSeeds, Inc., Lancaster, PA, USA, and 'Tetraprime' perennial ryegrass seed was provided by SeedWay, LLC, Penn Yan, NY, USA. The red clover and perennial ryegrass seeds were coated with different combinations of soy flour (SF), diatomaceous earth (DE), micronized vermicompost (MVC-2 and 3), and concentrated vermicompost extract (CVE) to identify the most stable and effective coating formulations (Table 1). Specific treatments and ratios of coating materials evaluated were (SF:DE 30:70, 40:60, 50:50 and 60:40, SF:MVC-2 (30:70), SF:MVC-3 (30:70), SF:DE:CVE (30:70)). A mechanical Ro-Tap shaker (Ro-Tap Testing Sieve Shaker No. 1506; The W.S. Tyler Co., Cleveland, OH) was utilized to sieve the SF to obtain a particle size smaller than (<75 µ), as previous studies have shown that smaller particle size results in more even distribution of the SF coating on seeds [15]. Seed coating biostimulant materials used in this research were previously analyzed by the Cornell Soil Health Nutrient Analysis Laboratories and recently reported in [27].

**Table 1.** Materials used as seed coating biostimulant treatment formulations in this study.

| Coating Materials | Abbreviation | Source |
|---|---|---|
| Soy Flour | SF | Archer Daniels Midland Co., Decatur, IL, USA |
| Diatomaceous Earth | DE | Perma-Guard, Inc., Albuquerque, NM, USA |
| Concentrated Vermicompost Extract (liquid) | CVE | Worm Power, Avon, NY, USA (concentrated by Caloris Engineering, Easton, MD, USA) |
| Micronized Vermicompost | MVC-2 | Worm Power, Avon, NY, USA |
| Micronized Vermicompost | MVC-3 | TerraVesco, Sonoma Valley Worm Farm, Sonoma, CA, USA |

### 2.2. Seed Coating

A 15-cm diameter, R-6 (Universal Coating Systems, Independence, OR, USA) laboratory-scale rotary pan coater was used to coat seeds in all experiments (Figure 1). Each seed coating treatment consisted of two components: dry powder and liquid. SF and DE, SF and MVC-2, and SF and MVC-3 were applied as dry powder to the seed surface with distilled water. For the SF, DE, and CVE treatments, water was replaced with liquid compost extract. For each treatment, the powder and liquid were applied to the surface of the seeds in incremental amounts as they rotated in the R6 to achieve uniform results. To clean the residual dust of each coating batch and avoid cross contamination of treatments, the R-6 pan was cleaned with a sponge and hot water-liquid soap solution. This was followed by cleaning with disinfectant wipes (three times), and R-6 with high pressure air flow was applied around the pan and cylinder to ensure completely drying. In this study, coating combinations of SF:DE and SF:MVC for both crops were applied on separate days.

Twenty-five grams of seed were used for red clover (1000 seed weight = 2.5 g) treatments, and 15 g of seed were used for the perennial ryegrass (1000 seed weight = 1.5 g). Therefore, we had an equal amount of treated seeds (~10,000 seeds) for each treatment of each crop. The total build up percentage was approximately 30% for clover and 70% for perennial ryegrass (Figure 2). Variation in the percent build up needed to achieve uniform coverage reflects the need for specifically developed seed coatings for each seed type and species. The size and shape of seeds influences uniformity of treatments on the seeds. After coating, the seeds were dried at room temperature for 24 h (h) until completely dry [15,28]. To improve observation of the seed coating uniformity of the SF:DE and SF:DE:CVE blend, a red dye (Pro-Ized red colorant, Bayer Corp, Research Triangle Park, NC, USA) was added to the binder (1.0 mL dye per 10 mL binder). Due to the dark brown color of the MVC materials, no dye was used for SF and MVC combinations; the natural color showed coating quality and uniformity of application.

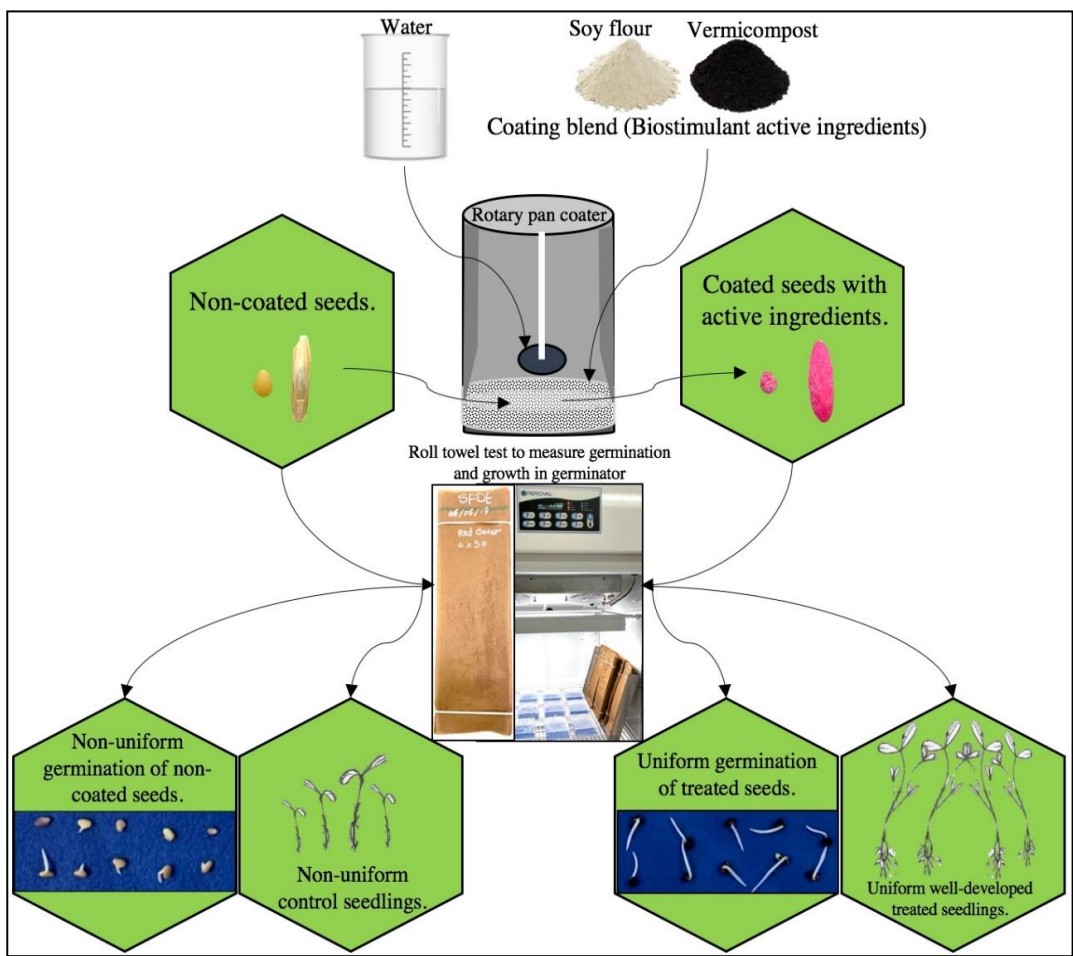

**Figure 1.** Figure of seed coating methodology used as an approach for application of biostimulant compounds for sustainable agriculture.

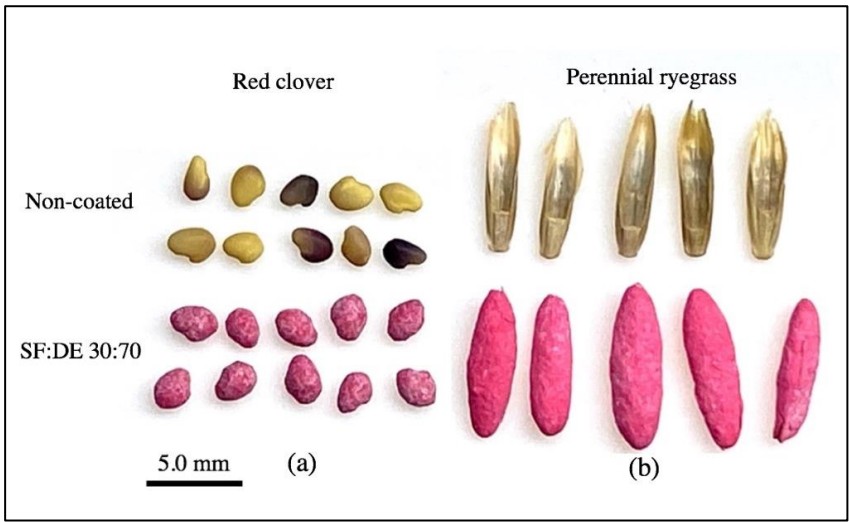

**Figure 2.** Non-coated and coated (SF:DE 30:70) red clover seeds with 30% build up (**a**) and non-coated and coated (SF:DE 30:70) perennial ryegrass with 70% build up (**b**). To improve observation of seed coating uniformity, a red dye was added to the binder.

### 2.3. Seed Coat Physical Properties

2.3.1. Seed Coating Integrity Test

The strength of the coating is an important quality as it relates to germination and potential for damage during handling, transportation, and planting. The surface material of coated seeds must have good mechanical properties to ensure that they do not crack or disintegrate before sowing. A Ro-Tap sieve shaker (The W.S. Tyler Co., Cleveland, OH, USA) was used to test the integrity of the coated seeds [15]. Four replications of 1.5 g of coated red clover seeds and four replications of 1.5 g of perennial ryegrass seeds from each coating formulation (treatments listed in Table 2) were tested to assure reliability and reproducibility. Samples were weighed and shaken for 2 min using a standard Ro-tap test shaker with U.S. Standard Testing Sieve No. 25 (0.71 mm opening) and a solid catch pan. Each sample was weighed again, and the percentage of coating loss was calculated according to the weight before and after the Ro-tap procedure. The weight of coating material, which passed through a No. 25 sieve was reported as weight loss (WL %) of material.

**Table 2.** Results of seed coating physical property testing, weight loss (WL, %), disintegration time (DT, min), compressive strength (Force N), time to break (TB) seed coating measured in s, relaxation time (RT) after the seed coating was fractured measured in s for coated seeds of red clover and perennial ryegrass.

| Crop | Treatment | WL (%) | DT (min) | Force (N) | TB (s) | RT (s) |
|---|---|---|---|---|---|---|
| | SF:DE 30:70 | 1.5 a * | 58 a | 16.2 a | 4.4 a | 0.3 a |
| | SF:DE 40:60 | 1.2 ab | 75 b | 19.2 b | 5.1 b | 0.36 b |
| | SF:DE 50:50 | 0.6 bc | 100 c | 20.6 b | 5.6 c | 0.48 c |
| Red clover | SF:DE 60:40 | 0.4 c | 103 c | 23.9 c | 5.5 c | 0.5 c |
| | SF:MVC-2 | 1.2 ab | 78 b | 16.8 a | 4.4 a | 0.38 b |
| | SF:MVC-3 | 1.1 ab | 83 b | 16.5 a | 4.7 a | 0.32 a |
| | SF:DE:CVE | 1.0 ab | 80 b | 17.5 b | 4.9 a | 0.31 a |
| | SF:DE 30:70 | 1.4 A * | 40 A | 15.3 A | 5.7 A | 0.2 A |
| | SF:DE 40:60 | 1.2 AB | 58 B | 17.9 B | 6.0 B | 0.38 B |
| Perennial ryegrass | SF:DE 50:50 | 0.5 B | 90 C | 20.9 C | 6.4 C | 0.51 C |
| | SF:MVC-2 | 1.3 B | 60 B | 15.6 A | 5.4 A | 0.24 A |
| | SF:MVC-3 | 0.9 BC | 55 B | 16.1 A | 5.5 A | 0.22 A |
| | SF:DE:CVE | 0.9 BC | 60 B | 15.8 A | 5.8 A | 0.31 B |

* Different letters within each column for each crop indicate significant differences using a Least Significant Difference (LSD) test at a significance level of $p < 0.05$. Lower case letters represent significant differences in red clover treatments and upper case letters denote perennial ryegrass treatment differences.

2.3.2. Mechanical Property Test

A texture analyzer (TA-XTplusC, Texture Technologies Corp., Hamilton, MA) was used to test the compressive strength of coated seeds. The TA-XTplusC is a precision instrument used to measure the surface mechanical properties of coated seeds and the compressive strength of a single seed. The arm of the texture analyzer containing a weighing sensor moves in a downward motion to compress the coated seed placed on the base of the analyzer and then returns to its original position. Data are assessed as the compressive strength (Force N) and time to breakage (TB, measured in seconds (s)) required to fracture the seed coating. The relaxation time (RT), which is the time required to completely rupture the seed coating was also measured [29,30]. After the seed coating was completely broken, the force (N) increases until the seed embryo was crushed (Figure 3). Texture analyzer software (Exponent Connect, version 7.0.2.0, S. Hamilton, MA, USA, 2018) was used to record force for TB and RT [26]. Ten coated seeds were randomly selected from batches of different formulations (SF:DE = 30:70, 40:60, 50:50 and 60:40, SF:MVC-2 (30:70), SF:MVC-3 (30:70), SF:DE:CVE (30:70)) to test their surface compressive strength for both red clover and perennial ryegrass coated seeds.

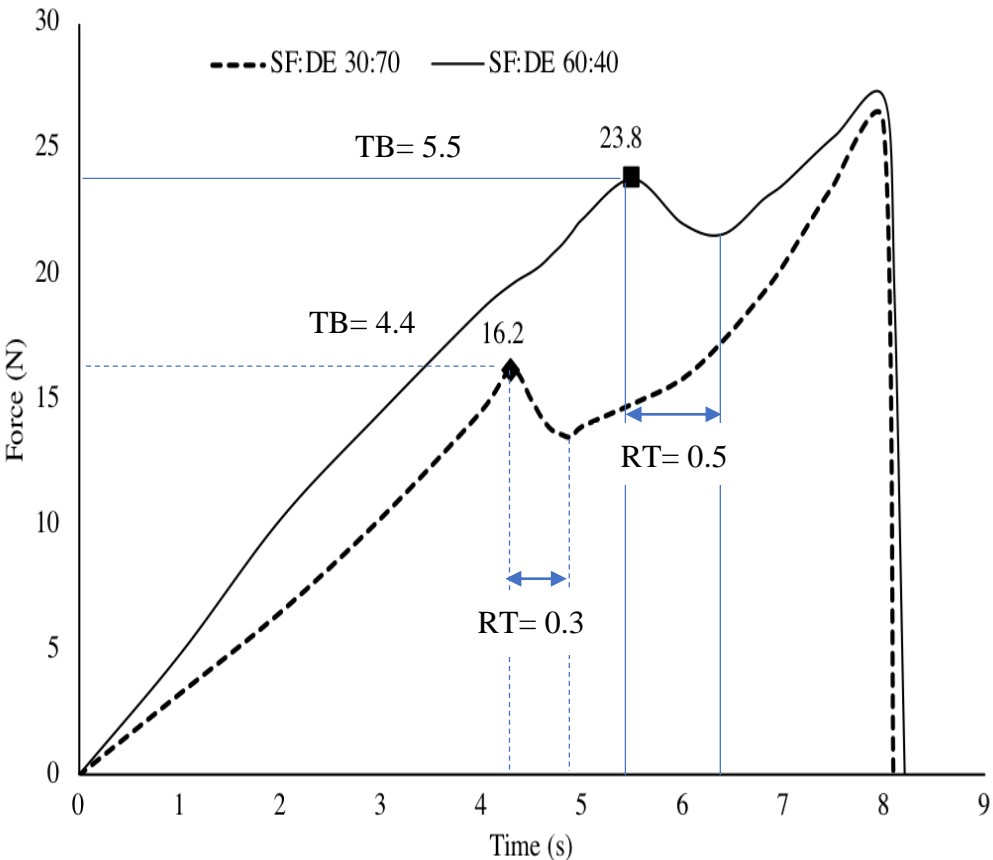

**Figure 3.** The values of peak load force required to break the seed coat of a single seed from two different seed coating blends of soy flour (SF) and diatomaceous earth (DE) tested at room temperature (SF:DE 30:70 and 60:40) for red clover are 16.2 and 23.8 N, respectively. The maximum force value (N) is a measure of coating strength and shows the maximum force needed to break the seed coat. Time to break seed coating (TB) and Relaxation Time (RT) after seed coat fracture until the coat is completely broken, both measured in s, are shown for a single seed. Force, TB, and RT data shown in Table 2 are the means of 10 seeds (Equipment: TA-XTplusC, Texture Technologies Corp., Hamilton, MA, USA, Software: Exponent Connect, version 7.0.2.0).

### 2.3.3. Seed Coating Hydration Test

The wet strength of a seed coating is largely dependent on the adhesion of the components after immersion in water. In theory, the slower the decomposition rate of coated seeds in water is, the more likely it is to delay germination. The hydration test was used to investigate the integrity of the coating materials when immersed in water. Hydration tests were conducted to determine the potential for seed coatings to prevent or delay germination. Four replicates of 100 coated seeds with different proportions of soy flour and diatomaceous earth, SF:DE (30:70, 40:60, 50:50 and 60:40), and the soy flour micronized vermicompost treatments SF: MVC-2, SF: MVC-3 were placed in 5 mL of distilled water to determine disintegration time. Disintegration time was measured in minutes.

### *2.4. Seed Germination and Seedling Growth Measurements*

Four replicates of 50 treated and non-coated control seeds were placed on two 30 cm × 45 cm moistened germination paper towels (Anchor Paper Company, St. Paul, MN, USA); then, an additional moistened standard germination paper towel was placed on top of the seeds. The towels were rolled and positioned in a germinator (Percival germinator, Model I-36LL, Perry, IA, USA). For perennial ryegrass seed, the germinator was maintained at 15/25 °C, with a 16/8 h photoperiod [31]; red clover, was maintained at a constant 20 °C with the same photoperiod [31]. Radical emergence (>2 mm) was

used to determine successful seed germination. The number of newly germinated seeds for both red clover and perennial ryegrass was recorded every 24 h. For perennial ryegrass, the total germination percentage (Gmax %) was recorded after 10 days. The Gmax % for red clover was recorded after seven days. Gmax %, the number of germinated seeds and germination uniformity (GU), (GU = time required for 90% germination subtracted by time required for 10% germination) were calculated [32] for each treatment. In addition, germination rate (T50, the time in h to reach 50% total germination) was calculated according to the equation developed by Coolbear et al. [33].

Root and shoot measurements (cm) were conducted in separate roll towel experiments (under the same growing conditions described above for seed germination) for each treatment using four replicates of 50 seeds. The seed vigor index (SVI) was equal to Gmax % multiplied by seedling length (combined root and shoot lengths) divided by 100 [34]. Seedlings were measured a week after full emergence for both crops. After measuring shoot and root lengths, all seedlings from each treatment were dried in an oven at 80 °C for 48 h to obtain the dry weight data.

## 2.5. Statistical Analysis

In all experiments, normality tests were conducted prior to ANOVA and all data passed the normal distribution test at a significance level of 0.05. Analysis of variance (ANOVA) ($\alpha$ = 0.05) and Fisher's least significant difference test for seed coating physical property and Dunnett test for germination and seedling growth data were performed on each of the significant variables measured by Minitab Express [35]. All Gmax % (Tables 2–4) and WL % (Table 2) data were arcsine transformed for analysis. Data for Gmax % and WL% are presented as non-transformed means (Tables 2–4). Pearson correlation was conducted for coating physical properties data using Minitab Express (Table 5).

**Table 3.** Germination and growth metrics of soy flour formulations as measured by total germination (Gmax %), germination rate (T50) measured in hours (h), germination uniformity (GU) measured in hours (h), shoot and root length (cm), and Seedling Vigor Index (SVI = Gmax % × Seedling length) of different coating formulas of SF:DE for red clover and perennial ryegrass.

| Crop | Treatment | Gmax (%) | T50 (h) | GU (h) | Shoot (cm) | Root (cm) | SVI |
|------|-----------|----------|---------|--------|------------|-----------|-----|
| Red clover | Control | 95 b * | 35 a | 37 a | 3.6 b | 2.4 b | 5.7 b |
| | SF:DE 30:70 | 98 a | 27 b | 27 b | 4.1 a | 3.0 a | 7.0 a |
| | SF:DE 40:60 | 99 a | 29 b | 25 b | 4.3 a | 2.9 a | 7.1 a |
| | SF:DE 50:50 | 96 b | 30 b | 27 b | 4.1 a | 2.9 a | 6.7 a |
| | SF:DE 60:40 | 96 b | 34 a | 28 b | 3.9 a | 2.7 a | 6.3 b |
| Perennial ryegrass | Control | 85 A * | 75 B | 39 B | 6.5 B | 5.5 B | 10.2 B |
| | SF:DE 30:70 | 83 A | 79 A | 40 B | 7.6 A | 6.2 A | 11.5 A |
| | SF:DE 40:60 | 83 A | 80 A | 42 B | 7.4 A | 6.3 A | 11.4 A |
| | SF:DE 50:50 | 80 B | 83 A | 47 A | 7.2 A | 5.9 A | 10.5 A |

* Different letters within each column for each crop indicate significant differences using a Dunnett test at a significance level of *p* < 0.05. Lower case letters represent significant differences between each red clover seed coating treatment compared with the control and upper case letters denote each of perennial ryegrass treatment differences compared with the control.

**Table 4.** Germination and growth metrics of soy flour/vermicompost formulations as measured by total germination (Gmax %), germination rate (T50) measured in hours (h), germination uniformity (GU) measured in hours (h), shoot and root length (cm), seedling dry weight (DW) recorded in grams (g), and Seedling Vigor Index (SVI = Gmax % × Seedling length) from evaluation of different coating materials for red clover and perennial ryegrass. The proportion of all coating materials is 30:70 (30% SF and 70% of DE or MVC).

| Crop | Treatment | Gmax (%) | T50 (h) | GU (h) | Shoot (cm) | Root (cm) | DW (g) | SVI |
|------|-----------|----------|---------|--------|-----------|-----------|--------|-----|
| | Control | 94 b * | 36 a | 35 a | 3.7 b | 2.5 b | 0.05 b | 6.0 b |
| | SF:DE | 98 a | 26 b | 27 b | 4.2 a | 2.8 a | 0.07 a | 6.9 a |
| Red clover | SF:MVC-2 | 99 a | 26 b | 24 b | 4.4 a | 2.9 a | 0.07 a | 7.3 a |
| | SF:MVC-3 | 99 a | 25 b | 25 b | 4.7 a | 3.0 a | 0.08 a | 7.6 a |
| | SF:DE:CVE | 98 a | 27 b | 25 b | 4.5 a | 3.2 a | 0.07 a | 7.5 a |
| | Control | 87 A * | 76 A | 41 B | 6.5 B | 5.9 B | 0.10 B | 10.8 B |
| | SF:DE | 85 A | 78 A | 43 B | 7.9 A | 6.5 A | 0.13 A | 12.3 A |
| Perennial ryegrass | SF:MVC-2 | 86 A | 77 A | 40 B | 8.1 A | 6.8 A | 0.14 A | 12.8 A |
| | SF:MVC-3 | 82 B | 81 A | 48 A | 8.3 A | 6.6 A | 0.13 A | 12.2 A |
| | SF:DE:CVE | 85 A | 77 A | 43 B | 8.4 A | 6.6 A | 0.15 A | 12.8 A |

* Different letters within each column for each crop indicate significant differences using a Dunnett test at a significance level of $p < 0.05$. Lower case letters represent significant differences between each red clover seed coating treatment compared with the control and upper case letters denote each of perennial ryegrass treatment differences compared with the control.

**Table 5.** Correlation coefficients between disintegration time (DT, min), weight loss (WL, %), and compressive strength (Force, N) from seed coating applications of soy flour and diatomaceous earth on red clover and perennial ryegrass seeds.

| Crop | | WL (%) | Force (N) |
|------|------|--------|-----------|
| Red clover | DT (min) | −0.99 *** | +0.92 ** |
| | WL (%) | - | −0.94 ** |
| Perennial ryegrass | DT (min) | −0.99 *** | +0.99 *** |
| | WL (%) | - | −0.96 *** |

**, *** Significant at $p < 0.001$, $0.0001$, respectively.

## 3. Results and Discussion

### 3.1. Seed Coating Physical Properties

The integrity and physical properties of coated and pelleted seeds are critical for overall performance. The production of dust can lead to health and environmental risks; therefore, it is important to quantitatively analyze the potential for breakage and weight loss that may occur during transportation and handling. In contrast, a seed coating that is too hard or impermeable to water may hinder germination. In this study, physical properties of the various seed coating formulations were tested by employing three different tests, including mechanical, texture, and hydration analyses.

Experimental results from the seed coating mechanical Ro-tap and Texture analysis (TA-XTplusC) tests are presented in Table 2 and Figure 3. For both crops, increasing the proportion of soy flour in the coating blend increased the compressive strength (Force N) of coated seeds. The time required to break the seed coating (TB), measured in s and relaxation time (RT) after breaking seed coating (Table 2) increased as soy flour proportion increased (Table 2). For example, for red clover, the average force (N) increased from 16.2 to 23.9 N as the soy flour content increased from 30% to 60%, which is an increase of approximately 48%. As soy flour content increased, the TB of coated seeds gradually increased from 4.4 to 5.5 s for red clover and 5.7 to 6.4 s for perennial ryegrass (Table 2). Although the same seed coating blend were used for both crops, the TB ranges were different most likely due to the difference in build-up percentage. There was a 1.1 s delay in breakage time for red clover when the content of soy flour increased from 30% to 60%, and 0.7 s delay for perennial ryegrass when the content

of soy flour increased from 30% to 50%. The force value to break down the coating for perennial ryegrass significantly increased by approximately 37% as soy flour increased from 30% (15.3) to 50% (20.9). Additionally, the weight loss of coated seeds from the Ro-tap test gradually decreased from 1.5% to 0.4% for red clover and 1.4% to 0.5% for perennial ryegrass (Table 2) as the proportion of soy flour in the seed coating increased. Interestingly, the mechanical properties of each seed formulation were not significantly different, even though the shape and surface properties of the seeds of both species differed.

There was no significant difference in both crops in terms of weight loss and disintegration time of coated seeds in water (Table 2) of soy flour and vermicompost. For red clover, when seed coatings with soy flour and vermicompost (30:70) (SF:MVC-2, SF:MVC-3 and SF:DE:CVE) were compared with SF:DE 30:70, the force (N) to break the seed coating increased by 0.6, 0.4, and 1.3 N, respectively and time to break (TB) the seed coating were non-significant, indicating that soy flour could serve as a binder for both types of materials used in this study (DE and vermicompost). There was no significant difference observed on force and TB data among treatments for perennial ryegrass (Table 2). Relaxation time after breaking red clover seed coating (RT) of SF:MVC-2, SF:MVC-3, and SF:DE:CVE increased slightly by 0.08, 0.02, and 0.01s, and the RT of perennial ryegrass increased by 0.04, 0.02, and 0.11s compared with that of SF:DE 30:70. In conclusion, the surface mechanical strength of different seed coating formulations with soy flour and vermicompost blends were non-significant but slightly higher than that of soy flour and diatomaceous earth.

The hydration test measures the time required to dissolve the coating materials. The proportion of soy flour in the seed coating blends had a significant effect on disintegration time (DT) (Table 2). A higher proportion of soy flour in the coating blend for red clover increased the DT from 58 to 103 min. This pattern was also observed for perennial ryegrass, as the proportion of soy flour ratio increased from 30% to 50%, the DT significantly increased from 40 to 90 min (Table 2). There were no significant differences in WL or DT for micronized vermicompost and compost extract (SF:MVC-2, SF:MVC-3 and SF:DE:CVE) seed treatments on either crop species (Table 2).

According to the American Seed Trade Association (ASTA), the key to a successful seed treatment is high physical integrity with low dust production. Determination of mechanical integrity of coated seeds is an important step in order to meet environmental safety standards [36]. Amirkhani et al. [15,27] tested the mechanical properties of several different broccoli coated seeds with Ro-tap and Texture Analyzer methods. The weight loss percentage that Amirkhani et al. [27] reported were slightly higher than the data collected from the red clover and perennial ryegrass seed treatments evaluated in this study for the same seed coating formulations. Total peak load force to break the seed coats for broccoli seeds were slightly lower than the value that was observed in this study for the SF:DE combination. Overall, the mechanical integrity of the coated red clover and perennial ryegrass seeds were more stable for the same seed coating formulation treatments, compared to broccoli. This difference might be because of the size and shape of clover and perennial ryegrass seeds contrasted to the broccoli seeds. Accinelli et al. [37] also attributed differences in seed dust emission for seed coating treatments of maize (*Zea mays* L.) and canola (*Brassica napus* L.) to seed physical characteristics. The mechanical integrity data observed for the different formulations in this study are in accordance with European Standards (Italy and France) and meet the benchmarks for safety of dust production of coated seeds [38].

*3.2. Germination and Seedling Growth of Soy Flour and Diatomaceous Earth Seed Coating*

In seed coatings with diatomaceous earth, soy flour served as the biostimulant component of the formulations. The results presented in Table 3 show that all coated treatments of red clover seeds significantly improved T50 and GU with no reduction in Gmax % compared to the non-treated control (Table 3), except for the SF:DE 60:40 treatment. Although soy flour proportions higher than 40% resulted in stronger and more durable seed coating mechanical properties, it had a negative effect on germination parameters (Gmax % and T50). For example, seeds treated with 30% and 40% soy flour had 98 and 99% Gmax % and T50 of 27 and 29 h, respectively, However, increasing the soy flour to

60% resulted in maximum germination of 96% and delayed the T50 to 34 h (Table 3). The negative effect in germination was attributed to the hard mechanical barrier of the seed coating with high soy flour content.

In contrast, for perennial ryegrass seeds, the Gmax % of all coated seeds was slightly reduced and showed delayed germination rates compared with the non-treated control. Control seeds of perennial ryegrass had the greatest Gmax % (85%) and significantly faster T50 (75 h) compared to all coating formulations. Application of 50% soy flour to the seed coating (SF:DE 50:50) reduced the Gmax % to 80% and T50 by 8 h and GU, compared with the non-treated control seeds. Due to the high percentage of coating build up (70%), the delay and a slight reduction in Gmax % was not unexpected. Several studies have indicated that the seed coating can act as a mechanical barrier for water absorbance and radical emergence [10,15].

Shoot and root length and seedling vigor index (SVI) are important indicators that determine whether the treated seeds promote seedling growth. Shoot and root length and seedling vigor index of treated seeds were significantly higher than those of non-treated control seeds for both crops (Table 3). The lowest application of soy flour (SF:DE 30:70) to the red clover seeds resulted in 4.1 cm shoot length, 3.0 cm root length, and 7.0 SVI, respectively, which were 14, 25, and 23 % higher than those of non-treated seeds. The same application rate of soy flour (SF:DE 30:70) to perennial ryegrass seeds improved the shoot growth by 17% and increased both root length and SVI values by approximately 13% compared to the control seeds. Amirkhani et al. [15,25–27] reported similar results for seed coatings that combined soy flour with diatomaceous earth. In their research, the seed coating blends had significant and positive effects on the above and below ground growth parameters of broccoli, tomato, radish, and hemp. They hypothesized that since soy flour is a plant-based protein and a rich source of several amino acids, it may have led to the increase in plant shoot and root growth and dry matter content and influenced uptake of nitrogen.

*3.3. Germination and Seedling Growth of Soy Flour and Vermicompost Seed Coating*

In addition to soy flour and diatomaceous earth, co-application of soy flour and vermicompost as rich sources of nutritional materials were tested as seed coatings and their effect on germination and seedling growth were recorded for red clover and perennial ryegrass. Shoot and root length, dry weight, and seedling vigor index of seedlings of all coated seed treatments were significantly higher compared to the non-treated controls (Table 4).

All treated red clover seeds germinated significantly faster (approximately 10 h) and had higher Gmax % (Table 4 and Figure 4) than non-coated seeds. They also germinated more uniformly than the non-treated control seeds. Gmax % was ≥ 98% for all treated seeds, which was significantly higher than control with 94% Gmax %. Red clover data showed that the shoot and root length and seedling vigor index of treated seeds were significantly higher than the non-treated control seeds. For example, compared with non-treated seeds control, the shoot length of SF:DE, SF:MVC-2, SF:MVC-3, and SF:DE:CVE increased by 14%, 19%, 27%, and 22%, respectively. Moreover, the root length of treated seeds increased by 12%, 16%, 20%, and 28%, respectively. All treatments showed a 40 to 60% increase in seedling dry weight (DW) compared with the control. The seedling vigor indexes (SVI) were 15%, 22%, 27%, and 25% higher than control, respectively (Table 4).

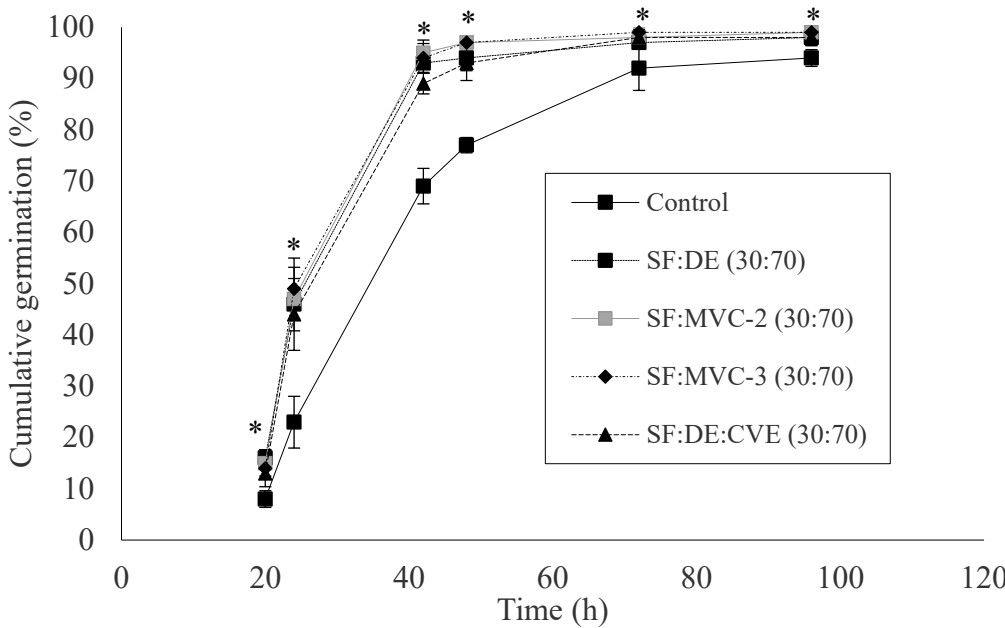

**Figure 4.** Cumulative germination percentage of red clover non-treated control seeds versus biostimulant coated seeds. * Significant at $p \leq 0.05$.

For perennial ryegrass, application of soy flour (SF:DE) and co-application with vermicomposts (SF:MVC-2 and 3, and SF:DE:CVE) increased shoot length by 22%, 25%, 28%, and 29% and root length by 10%, 15%, 12%, and 12%, respectively, compared to the non-treated control. The highest root length was observed in the SF:MVC-2 treatment. All treatments had higher DW than the control. Additionally, the highest SVI was observed in the SF:DE:CVE treatment, which was approximately 40% higher than the SVI of non-treated control (Table 4).

Statistical analysis (Pearson's correlation) of seed coating formulations and germination showed significant negative correlations between seed coating WL (%) and DT (min) (r = −0.99 ***). There was also a significant positive correlation between DT (min) and force (N) (r = +0.92 **). A significant negative correlation between WL (%) and force (N) (r = −0.94 **) from SF:DE coating formulations of red clover evaluated was observed (Table 5). For perennial ryegrass, the significant correlation coefficient between WL (%) and DT (min) was r = −0.99 *** and the correlation between WL (%) and force (N) was r = −0.96 ***. Lastly, for perennial ryegrass, the correlation between DT (min) and force (N) was r = +0.99 *** (Table 5). These data indicate that a higher proportion of SF in the seed coating formulation resulted in harder coatings but had only a slight impact on the Gmax %. For red clover, increasing the soy flour from 30% to 60% in seed coating formula reduced the Gmax % by 2%; however, T50 was significantly delayed by 7 h (Table 3). Similarly, for perennial ryegrass, increasing soy flour from 30% to 50% in the seed coating resulted in a 3% reduction in Gmax % and a minor delay on T50 (4 h).

In the present study, the seedling growth data for both cover crops evaluated indicate that seed coating can be an efficient and effective delivery method for application of nutritional biostimulant materials at the time of sowing for rangeland and grassland restoration. Several previous studies showed that applications of plant-based proteins and vermicompost can improve biometric growth parameters, related to production and yield of horticultural, field, and cover crops. Karlsons et al. [39] showed that a 10% addition of vermicompost in pure sand significantly increased fresh and dry weight of winter rye shoots by 578% and 265%, respectively. Tognetti et al. [40] found that application of vermicompost to degraded volcanic soil (to the extent of 20 and 40 g/kg soil) sown with ryegrass (*L. perene*) significantly increased ryegrass yields compared to the control due to the large nutrient concentrations and high microbial populations, when mixed with the soil. The positive effect of

vermicompost on plant growth in this study agrees with the results of Alwaneen [41] on alfalfa and Amirkhani et al. [27] on broccoli. Amirkhani et al. [27] found that dairy manure-based vermicompost can supply essential nutrients to plants to enhance growth as well as increase the organic matter contents of soil for higher crop production. Moreover, in the recent decade, several researchers have been working on treating plants with biostimulants to stimulate crop productivity and increase stress tolerance under dynamic abiotic stresses [42–44]. The cover crop seeds treated with biostimulants in combination with other bio-effectors, such as superabsorbent polymers to investigate the response of these plants to drought, can be an area of future studies.

## 4. Conclusions

Seed coating technology can be an effective strategy to maximize early stand establishment of cover crops. Biostimulants applied as seed coatings have the potential to effectively promote seedling growth, and early stand establishment of red clover and perennial ryegrass. In this study, biostimulant seed coatings promoted the seedling growth of red clover and perennial ryegrass seeds and accelerated the germination of red clover compared to the non-treated control seeds. More rapid germination could aid in establishment under arid conditions and in areas with poor soils. However, further studies are needed to determine if vermicompost and plant-based proteins can be developed for economical commercial applications as seed treatments. The seed industry commonly includes fertilizers and *Rhizobium*, nitrogen fixing bacterial inoculants for red clover seeding. Additional research will be needed to determine if the biostimulant materials used in these experiments are compatible with seed inoculants. The use of biostimulants in combination with vermicompost and other biofertilizers as seed coatings may offer a great opportunity to increase plant production and long-term sustainability in agricultural landscapes.

**Author Contributions:** Y.Q., M.A., and H.M. contributed equally in writing the manuscript. Z.C. was involved in reviewing the manuscript. A.G.T. assisted with conceptualization of the overall experiments, funding, writing, and reviewing the manuscript. All authors have read and agreed to the published version of the manuscript.

**Funding:** This material is based upon work that is supported by the National Institute of Food & Agriculture, US Department of Agriculture, Multi-state Project W-3168, under accession #1007938. Y.Q. was sponsored by China Scholarship Council.

**Acknowledgments:** The authors appreciate the technical assistance of Michael Loos in laboratory experiments and Zhen Wang for assistance with this project.

**Conflicts of Interest:** The authors declare that the research was conducted without any financial relationships that could be construed as a potential conflict of interest.

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
