# Peer review of "Biostimulant Seed Coating Treatments to Improve Cover Crop Germination and Seedling Growth"

_agronomy, doi:10.3390/agronomy10020154_

Round 1

Reviewer 1 Report

The paper was comparing the different combinations of biostimulant seed coating treatment on ryegrass and red clover, also tested for their seed coat physical properties, germination and shoot/root growth. In general, the paper is well written and understandable, but it is sometimes difficult to follow especially treatment structures that were not clear. 

Overall, here are some of the concerns since there was insufficient detailed information about the materials and methods to allow replication. Experiment design and treatment structures were not clear. It was not a full factorial of the combination of the dry or liquid treatments. Here are the reasons why the treatment structures are not clear: 

Coating integrity test – 4 replications and 1.5 grams of clover and 1.5 grams ryegrass seeds without treatment structures to measure the weight loss (WL%) but it was not mentioned what are the treatments.  Mechanical property test – assumed 10 replications because 10 seeds were tested, Force N, TB and RT were measured (or calculated) with 8 different combination formula: (SF:DE=0:100, 30:70, 40:60, 50:50 and 60:40, SF:MVC-2 (30:70), SF:MVC-3 (30:70), SF:DE:CVE (30:70)) (stated in line 153) but no data was presented in treatment 0:100 in tables 2 – 4 .  Coating hydration test – 4 replications with 100 seeds and disintegration time (DT) was measured in 7 combination formula: SF: DE (0:100, 30:70, 40:60, 50:50 and 60:40), and the soy flour micronized vermicompost treatments SF: MVC-2, SF: MVC-3. 

Also, there was inconsistent writing of abbreviations, such as hours – h, days – d, second – s, etc, throughout the article. The authors created abbreviation that was not used in the tables/figures, such as T90, T10, SL, etc. but there are some of the abbreviations that mentioned throughout the article except at the results and discussion section, it makes it harder to follow. Please be consistent with the writing. 

Here are some detailed comments: 

Introduction: 

Line 45: Please provide the citation of perennial ryegrass grown in southern Europe, the Middle East, Central Asian and North Africa. Line 47: please provide the citation for clovers that were grown to provide nitrogen and aid in weed suppression.  Line 48: Why only red clover in the treatment structure but not white clover? Were there any studies about white clover that can be included in the introduction section? Was planting cover crop a common practice in China?  Line 50: Please provide the citation of the benefits of cover cropping.  Line 56: please provide the citation for the benefits of seed enhancements.  Line 66: How was the seed coating for cover crop important in China? How is biostimulant seed coating different than the one in China’s market now? The authors did not address these issues to indicate how important the biostimulant seed coating in cover crops. Line 75: Please provide the citation of these coating/treatment procedures. 

Materials and methods:

Line 91: How does coating for broccoli and tomato related to this paper? There were not the same species and functions in the cropping system.  Line 98: Please provide the variety of red clover.  Line 97: It does not seem to be a full factorial combination. Please provide the detailed treatment structure of different combinations of the coating materials.  Line 106: Please provide the cleaning procedure of the R-6 rotary pan between different coating. Line 130: were clover seed and ryegrass tested separately?  Line 142: were the 10 coated seeds randomly selected batches of different formulations for both clover and ryegrass seeds?  Line 157: The sentence starts with ‘randomly selected from batches…’, it seems incomplete.  Line 180: Please provide the size of the paper towels.  Line 194: What was the growth stage when the measurements of root and shoot? 

Statistical Analysis

Line 200: The data analysis section was incomplete. I am not familiar with the software (Minitab express) but there was not enough information regarding the data analysis. For instance, were outliers, homogeneity of variance determined? Some of the data were transformed to perform mean separation (LSD) but it was not clear which dataset was transformed. Were replications tested as a random or fixed effect? The Pearson Correlation was conducted, and the result was not presented in a table or any form, however, the results were written, and the meaning of the asterisk symbols was not mentioned in the text. 

Results and discussion: 

Line 242: the weight loss was not mentioned tested in the water in any of the materials and methods section. Please clarify the methodology. Line 253 – 255: This should be in the materials and methods section. Line 265: The discussed seems to focus on the paper (citation no. 13, 22, 23, and 24) in broccoli, I suggest finding some articles of small grain seeds for better comparison. Line273: Please provide the citation of the European Standards for the safety of dust production Table 3: Why the combination of 60:40 of SF: DE was not on perennial ryegrass? Why control was not one of the treatment comparisons?  Table 4: It seems will be appropriate to do orthogonal contrast (or Dunnett test to compare with the control treatment) on table 3 and 4 (combined) since it is not a full factorial combination of the treatment, it is hard to analyze the treatment structure.  Figure 4: There was no statistical analysis in this figure, or any mean separation or standard error bars. Out of all the treatments, why there were only 2 treatments with control?  Line 350: Pearson’s correlation was not present in the table or figure and no description of the asterisk in the text.  Line 351: It should be WL% and DT for weight loss percentage and disintegration time.  Line 365: Please provide the citation of several previous studies. 

Author Response

Response to Reviewer 1

Point 1: “The paper was comparing the different combinations of biostimulant seed coating treatment on ryegrass and red clover, also tested for their seed coat physical properties, germination and shoot/root growth. In general, the paper is well written and understandable, but it is sometimes difficult to follow especially treatment structures that were not clear.”

Response 1:Authors of this manuscript would like to thank the Reviewer #1 for careful and detailed reading of our manuscript and for the thoughtful comments and constructive suggestions, which help to improve the quality of our manuscript. Our response (in red) to the general and specific comments follows (the reviewer’s points are in italics”).

General comments:

Overall, here are some of the concerns since there was insufficient detailed information about the materials and methods to allow replication. Experiment design and treatment structures were not clear. It was not a full factorial of the combination of the dry or liquid treatments. Here are the reasons why the treatment structures are not clear:

Point 1: “Coating integrity test – 4 replications and 1.5 grams of clover and 1.5 grams ryegrass seeds without treatment structures to measure the weight loss (WL%) but it was not mentioned what are the treatments.  Mechanical property test – assumed 10 replications because 10 seeds were tested, Force N, TB and RT were measured (or calculated) with 8 different combination formula: (SF:DE=0:100, 30:70, 40:60, 50:50 and 60:40, SF:MVC-2 (30:70), SF:MVC-3 (30:70), SF:DE:CVE (30:70)) (stated in line 153) but no data was presented in treatment 0:100 in tables 2 – 4.  Coating hydration test – 4 replications with 100 seeds and disintegration time (DT) was measured in 7 combination formula: SF: DE (0:100, 30:70, 40:60, 50:50 and 60:40), and the soy flour micronized vermicompost treatments SF: MVC-2, SF: MVC-3.” 

Response 1: Thanks for bringing this issue to our attention. We attempted to make a coating with no SF (SF:DE 0:100) but because of the lack ofbinding effectthe coating materials around the seed were extremely fragile and before performing mechanical tests they were falling apart. To avoid confusion authors decided to delete SF:DE 0:100 from the treatments list mentioned on the manuscript M&M. We added a statement in the materials and methods [Line 113-114] to add clarity to the treatments evaluated.

The following is the edited sentence on coating integrity test:

“Four replications of 1.5 g of coated red clover seeds and four replications of 1.5 g of perennial ryegrass seeds from each coating formulation (treatments listed in Table 2) were tested to assure reliability and reproducibility.”

Point 2: “Also, there was inconsistent writing of abbreviations, such as hours – h, days – d, second – s, etc, throughout the article. The authors created abbreviation that was not used in the tables/figures, such as T90, T10, SL, etc. but there are some of the abbreviations that mentioned throughout the article except at the results and discussion section, it makes it harder to follow. Please be consistent with the writing.”

Response 2: On writing abbreviations authors accepted the reviewer comments and throughout the manuscript “h” used for hours, “s” used for seconds, “T50” used for “germination rate”, and “Gmax%” used for “Total germination percentage” and “total germination”. T90, T10 and SL were deleted.

Specific comments:

Introduction: 

Point 1: “Line 45: Please provide the citation of perennial ryegrass grown in southern Europe, the Middle East, Central Asian and North Africa.”

Response 1: The following citation added to the line 56.

“Balfourier, F.; Imbert, C.; Charmet, G. Evidence for phylogeographic structure in Lolium species related to the spread of agriculture in Europe. A cpDNA study. Theoretical and Applied Genetics, 2000101(1-2), 131-138.”

Point 2: “Line 47: please provide the citation for clovers that were grown to provide nitrogen and aid in weed suppression.” 

Response 2: The following citation added to the line 59.

“McKenna, P.; Cannon, N.; Conway, J.; Dooley, J., Davies, W.P. Red clover (Trifolium pratense) in conservation agriculture: a compelling case for increased adoption. International journal of agricultural sustainability, 2018 16(4-5), 342-366.”

Point 3: “Line 48: Why only red clover in the treatment structure but not white clover? Were there any studies about white clover that can be included in the introduction section? Was planting cover crop a common practice in China?” 

Response 3: We were interested in determining the influence on seed coatings on a clover species. Since white clover seeds are very small the authors decided to test red clover to determine if there was a treatment effect. Since the focus was not on white clover we did not include references for white clover seed treatments.

Use of cover crops and green manures are common practices in China, however, crop establishment in poor soil and arid conditions has been a problem.

Point 4: “Line 50: Please provide the citation of the benefits of cover cropping.” 

Response 4: The following citation added to the line 62.

“Daryanto, S.; Fu, B.; Wang, L.; Jacinthe, P.A.; Zhao, W. Quantitative synthesis on the ecosystem services of cover crops. Earth-Science Reviews, 2018,185,357-373.”

Point 5: (a)“Line 56: Please provide the citation for the benefits of seed enhancements. (b) Line 66: How was the seed coating for cover crop important in China? (c) How is biostimulant seed coating different than the one in China’s market now? The authors did not address these issues to indicate how important the biostimulant seed coating in cover crops.”

Response 5 (a): The following citation added to the line 68.

“Taylor, A.G.; Allen P.S.; Bennett, M.A.; Bradford K.J.; J.S. Burris, J.S.; Misra, M.K. Seed enhancements. Seed Sci. Res.1998, 8, 245–256.”

Response 5 (b): Additional information has been provided (Line 74-77).

The following three research and review articles performed in China indicating the importance of seed coating technology for grassland restoration in semi-arid areas of China such as Inner Mongolia state.

“He, Z.X.; Mao, P.S.; Sun, Y.; Li, M. Review of grass seed coating technology. Acta Agrestia Sinica 2016, 2, 270–277. (Chinese with English abstract)”

“Li, C.Y.; Zhang, F.; Liu C.H.; Yu, F.S.; Li, Y.C. Screening of coating materials for forage seeds. Journal of Northeast Agricultural University2013 4, 94–100.(in Chinese with English abstract)

“Ou, C. M.; Mao, P.S. Progress in research and application of coating technology for grasses.Seed. 2019 11, 63–67.” (in Chinese)

Response 5 (c): The authors are not aware of any biostimulant seed treatments on the market in China. Very little research has been conducted on plant-based biostimulants (such as soy flour) or nutritional materials (such as vermicompost) as  seed treatments.

Point 6: “Line 75: Please provide the citation of these coating/treatment procedures.” 

Response 6: Citations were added to the line 88.

Materials and Methods:

Point 1: “Line 91: How does coating for broccoli and tomato related to this paper? There were not the same species and functions in the cropping system.” 

Response 1: There are several reasons why authors referred to the coating delivery system developed for broccoli and tomato. To develop the delivery system of treatments using seed coating technology, the method of application and the performance of the binder and filler (with particular particle sizes) to reach to the desirable build-up of materials around the seed are critical and important. The protocol developed in “Ref 15, 26-28” during 2014 to 2019 were the basis for the initiation of this research project.

Point 2: “Line 98: Please provide the variety of red clover.” 

Response 2: The variety of red clover was not stated and this section has been corrected as follow:

Line 108: “Red clover ‘VNS’- variety not stated seed was obtained from King’s AgriSeeds …”

Point 3: “Line 97: It does not seem to be a full factorial combination. Please provide the detailed treatment structure of different combinations of the coating materials.” 

Response 3:This sentence was added (line 113-114) for clarity.

“Specific treatments and ratios of coating materials evaluated were (SF:DE 30:70, 40:60, 50:50 and 60:40, SF:MVC-2 (30:70), SF:MVC-3 (30:70), SF:DE:CVE (30:70)).”

Point 4: “Line 106: Please provide the cleaning procedure of the R-6 rotary pan between different coating.”

Response 4: Information has been added to lines 128-132 as follows:

“To clean the residual dust of each coating batch and avoid cross contamination of treatments, the pan of the R-6 was cleaned with a sponge and hot water-liquid soap solution. This was followed by cleaning with disinfectant wipes (3 times) then totally dried around the pan and cylinder of R-6 with high pressure air flow (In this study coating combinations of SF:DE and SF:MVC for both crops were applied on separate days.).”

Point 5: “Line 130: were clover seed and ryegrass tested separately?” 

Response 5: Integrity, mechanical property, and hydration tests for red clover and ryegrass were performed separately on the same day.

Point 6: “Line 142: were the 10 coated seeds randomly selected batches of different formulations for both clover and ryegrass seeds?”

Response 6: Yes for both crops,  10 coated seeds were selected randomly. The following change has been made for line 178 to 180:

“Ten coated seeds were randomly selected from batches of different formulations (SF:DE= 30:70, 40:60, 50:50 and 60:40, SF:MVC-2 (30:70), SF:MVC-3 (30:70), SF:DE:CVE (30:70)) to test their surface compressive strength for both red clover and perennial ryegrass coated seeds.”

Point 7: “Line 157: The sentence starts with ‘randomly selected from batches…’, it seems incomplete.” 

Response 7: This has been edited. Please see Line 178 to 180.

Point 8: “Line 180: Please provide the size of the paper towels.”

Response 8: The dimension of paper towel added to line 206:

“30 cm ´45 cm moistened germination paper towels” 

Point 9: “Line 194: What was the growth stage when the measurements of root and shoot?”

Response 9: The growth stage of shoot and root measurements were “seedling stage”. The following statement was added to line 224-225:

“Seedlings were measured a week after full emergence for both crops.”

Statistical Analysis

Point 1: “Line 200: The data analysis section was incomplete. I am not familiar with the software (Minitab express) but there was not enough information regarding the data analysis. For instance, were outliers, homogeneity of variance determined? Some of the data were transformed to perform mean separation (LSD) but it was not clear which dataset was transformed. Were replications tested as a random or fixed effect? The Pearson Correlation was conducted, and the result was not presented in a table or any form, however, the results were written, and the meaning of the asterisk symbols was not mentioned in the text.”

Response 1: To clarify the statistical analysis, the following information is provided and changes have been made:

“I am not familiar with the software (Minitab express) but there was not enough information regarding the data analysis. For instance, were outliers, homogeneity of variance determined?”

In all versions of MINITAB under STAT>ANOVA>Test for Equal variance “Levene Test”, we can determine whether the variances or standard deviations of two or more groups differ. Authors made sure that in datasets of this study the p. value of Levene test were > 0.05and data had equal variances therefore homogeneity of variances were not violated.

“Some of the data were transformed to perform mean separation (LSD) but it was not clear which dataset was transformed.”

Line 230-233: “All Gmax% (Table 2-4) and WL% (Table 2) data were arcsine transformed for analysis. Data for Gmax% and WL% are presented as non-transformed means (Table 2-4).”

“Were replications tested as a random or fixed effect?”

For this type of study replications were tested as a fixed effect since this study focused on individual effects of different treatments or coating formulations on entire planted seeds in each replicate.

“The Pearson Correlation was conducted, and the result was not presented in a table or any form, however, the results were written, and the meaning of the asterisk symbols was not mentioned in the text.”

Authors agree with the reviewer #1 to present the correlation data as Table 5. Please see line 458-461:

Table 5. Correlation coefficients between disintegration time (DT, min), weight loss (WL, %), and compressive strength (Force, N) from seed coating applications of soy flour and diatomaceous earth on red clover and perennial ryegrass seeds.

Crop

WL (%)

Force (N)

Red clover

DT (min)

-0.99***

+0.92**

WL (%)

-

-0.94**

Perennial ryegrass

DT (min)

-0.99***

+0.99***

WL (%)

-

-0.96***

**, *** Significant at p < 0.001, 0.0001, respectively.

Results and discussion: 

Point 1: “Line 242: the weight loss was not mentioned tested in the water in any of the materials and methods section. Please clarify the methodology.”

Response 1: In section “2.3.1. Seed Coating Integrity Test” we explained in detail [Line 156-166] the method of determination of WL%. This test was performed with dry seeds.

Point 2: “Line 253 – 255: This should be in the materials and methods section.”

Response 2: This change has been made. Please check lines 199-200 of M&M and line 300 of R&D.

Point 3: “Line 265: The discussed seems to focus on the paper (citation no. 13, 22, 23, and 24) in broccoli, I suggest finding some articles of small grain seeds for better comparison.”

Response 3: We have added an additional reference related to seed coating emissions for corn and canola [Please check Line 323-325].

 “Accinelli, C.; Abbas, H.K.; Little, N.S.; Kotowicz, J.K.; Mencarelli, M.; Shier, W.T. A liquid bioplastic formulation for film coating of agronomic seeds. Crop Protection, 2016 89, pp.123-128.”

Point 4: “Line273: Please provide the citation of the European Standards for the safety of dust production.”

Response 4: Authors provided the following citation that shows the Eu standards (Italy/France) on “Figure 1 Foqué et al. 2014”. Please check line 326-327.

“Foqué, D.; Devarrewaere,W.; Verboven, P.; Nuyttens, D. Physical and chemical characteristics of abraded seed coating particles. Asp. Appl. Biol. 2014, 122, 85–94.”

Point 5: “Table 3: Why the combination of 60:40 of SF: DE was not on perennial ryegrass?”

Response 5: The highest amount of soy flour possible to add to the coating blend for ryegrass was 50%, authors tested SF:DE 60:40 blend for ryegrass. Because of the shape and sharp edges of these seeds they do not rotate and tumble as well as round or kidney shape seeds in the pan and therefore stick to each other in the pan and wall of the cylinder of the R-6.

Point 6: “Why control was not one of the treatment comparisons?”

Response 6: In both “Tables 3 and 4” control was included and was one of our treatments.

Point 7:“Table 4: It seems will be appropriate to do orthogonal contrast (or Dunnett test to compare with the control treatment) on table 3 and 4 (combined) since it is not a full factorial combination of the treatment, it is hard to analyze the treatment structure.” 

Response 7: Authors performed Dunnett test on data in tables 2 and 3 and changes have been made throughout the manuscript. [Please find line 230-231 of Statistical Analysis and Table 3 (line 347-354) and Table 4 (line 390-399) of R&D].

The experiments and treatments were not designed for factorial analysis. The variables in table 2 and 3 were not continues and experiments were performed at different times therefore treatment combinations are more conducive to be analyzed separately. In this paper, the purpose of using LSD Fisher test was to compare any of treated seeds to each other and to the control, however Dunnett test is also appropriate and can determine the difference between each coating treatment and the control.

Point 8: Figure 4: “There was no statistical analysis in this figure, or any mean separation or standard error bars. Out of all the treatments, why there were only 2 treatments with control?”

Response 8: Authors agree with the reviewer #1 comment on figure 4 and a new Figure 4 of cumulative graph of germination % was added to illustrate the enhancement of red clover germination of coated seeds compared to the non-coated control seeds [Line 421-442].

Figure 4. Cumulative germination percentage of red clover non-treated control seeds versus biostimulant coated seeds. * significant at p £0.05.

Point 9: “Line 350: Pearson’s correlation was not present in the table or figure and no description of the asterisk in the text.”

Response 9: Authors agree with reviewer #1 and changes have been made to address this issue. Please see line 458-462.

Point 10: “Line 351: It should be WL% and DT for weight loss percentage and disintegration time.” 

Response 10: This has been changed. Please see line 445-447.

Point 11: Line 365: Please provide the citation of several previous studies.

Response 11: Additional citations have been added to the references section.

Reviewer 2 Report

In the manuscript presented “Biostimulant seed coating treatments to improve cover crop germination and seedling growth”  the authors explore the effects of different coating treatments (soy flour, diatomaceous earth, concentrated vermicompost extract, micronized vermicompost) on seed germination parameters, seedling growth and seed mechanical properties. The aim of the study is in line with the scope of this journal and the research is of interest for its agronomical information.

The work is well written and the results are clearly presented and discussed.

I have just find minor mistakes in the abstract and I have added some suggestions:

Line 12-13: the sentence can be stop after perennial ryegrass seeds (add the plural form of seed). The part of the sentence “as a tool to enhance germination and growth for cover crop establishment” can be removed because it is redundant.

Line 17: substitute “and for” with “whereas”.

Line 25: “on” should be substituted with “of” seeds.

Line 29: the subject of the sentence is “seed coating technologies”, so the verb “has” should be changed in “have”.

Line 34-35: This first sentence is unclear, please rephrase.

Figure 2: the addition of the red dye is mentioned only in the figure and not in the text. I suggest to add some details on this dye (chemical name, the brand etc) and its addition to the coating treatment in the text (line 120). In the figure it should be specified the coating material (using the abbreviations (SF, DE etc)) near the picture of each coated seed.

Line 187-189: “For perennial ryegrass, ....” this information should be moved in results and discussion.

Line 313-315: this is a interesting hypothesis and it could be important to monitor, in the future, the photosynthetic and growth performances of these seedlings for some weeks after being transplanted in the soil.

Author Response

Response to Reviewer 2

Point 1: “In the manuscript presented “Biostimulant seed coating treatments to improve cover crop germination and seedling growth”  the authors explore the effects of different coating treatments (soy flour, diatomaceous earth, concentrated vermicompost extract, micronized vermicompost) on seed germination parameters, seedling growth and seed mechanical properties. The aim of the study is in line with the scope of this journal and the research is of interest for its agronomical information. The work is well written and the results are clearly presented and discussed.

Response 1:The authors thank Reviewer #2for valuable and positive feedback on this manuscript.

Point 2: “Line 12-13: the sentence can be stop after perennial ryegrass seeds (add the plural form of seed). The part of the sentence “as a tool to enhance germination and growth for cover crop establishment” can be removed because it is redundant.”

Response 2: This has been corrected.

Point 3: “Line 17: substitute “and for” with “whereas”.”

Response 3: This has been corrected.

Point 4: “Line 25: “on” should be substituted with “of” seeds.”

Response 4: This has been corrected [Line 24].

Point 5: “Line 29: the subject of the sentence is “seed coating technologies”, so the verb “has” should be changed in “have”.”

Response 5: This has been corrected [Line 28].

Point 6: “Line 34-35: This first sentence is unclear, please rephrase.”

Response 6: This has been corrected as follows [Line 33-35]: “Exponential growth in the human global population, from 1.7 billion in 1900 to approximately 7.6 billion in 2019, has led to the over use and degradation of agricultural landscapes including grasslands used for grazing, forage and food production [1,2].”

Point 7: “Figure 2: the addition of the red dye is mentioned only in the figure and not in the text. I suggest to add some details on this dye (chemical name, the brand etc) and its addition to the coating treatment in the text (line 120). In the figure it should be specified the coating material (using the abbreviations (SF, DE etc)) near the picture of each coated seed.”

Response 7: Caption of “Figure 2” has been edited and “coating formula and crop names were added to the new Figure 2”. 

The following statement was added to the end of section 2.2. Seed Coatingof M&M [Line 154-158]:

“To improve observation of the seed coating uniformity of SF:DE and SF:DE:CVE blend a red dye (Pro-Ized red colorant, Bayer Corp, Research Triangle Park, NC, USA) was added to the binder (1.0 ml dye per 10ml binder). Due to the natural dark brown color of MVC materials, no dye was used for SF and MVC combinations, the natural color showed the coating quality and uniformity of application.”

Figure 2. Non-coated and coated (SF:DE 30:70) red clover seeds with 30% build up (left) and non-coated and coated (SF:DE 30:70) perennial ryegrass with 70% build up (right). To improve observation of the seed coating uniformity a red dye was added to the binder.

Point 8:“Line 187-189: “For perennial ryegrass, ....” this information should be moved in results and discussion.”

Response 8: With respect to the reviewer #2 opinion, authors of this manuscript believe that the following statement [line 224-225] is part of the standard method of recording germination data of red clover and perennial ryegrassand it belongs in the M&M section.

“For perennial ryegrass, the maximum percent germination (Gmax%) was recorded after 10days. The Gmax% for red cloverwas recorded after 7 days.”

Point 9:“Line 313-315: this is an interesting hypothesis and it could be important to monitor, in the future, the photosynthetic and growth performances of these seedlings for some weeks after being transplanted in the soil.”

Response 9: This suggestion of reviewer #2 will be considered for the future step of this research on cover crop seed coating with biostimulant materials.  

Round 2

Reviewer 1 Report

The authors have provided significant clarification/explanation and detailed information in the response/Material and Methods section for the biostimulant seed coating treatment paper.  They have also provided more citations to support the results and added a few tables and figures to clarify the results.